# Drought stress induces salicylic acid accumulation, altering monoterpene profile and suppressing resin duct formation in Engelmann spruce

**Thomas Seth Davis** [ID]\*, **Ehsan Khedive, Edward Hill, Troy Ocheltree**

Forest & Rangeland Stewardship, Warner College of Natural Resources, Colorado State University, Fort Collins, Colorado, United States of America

\* seth.davis@colostate.edu

## Abstract

Drought is a critical stressor on plants and often precedes large-scale insect outbreaks in forest ecosystems. Whether plant physiological mechanisms underlie this pattern remains uncertain; environmental conditions affect the growth rate of insect populations but also have consequences for plant defense phenotypes. To investigate the latter, we experimentally applied water stress to test relationships between drought, salicylic acid (SA) accumulation, and tree resistance factors including secondary metabolite (monoterpene) concentrations and formation of traumatic resin ducts in Engelmann spruce (*Picea engelmannii*), a widespread forest tree in western North America. Three key findings emerged. First, both acute and chronic water stress reduced stem water potentials, with evidence for loss of photosynthetic function as water potentials declined below −2.0 MPa. Negative water potential was associated with an increase in the concentrations of SA in both needles and stem phloem, but this trend was stronger in needles. Second, under water stress elevated SA concentrations were associated with increased concentrations of several structurally similar monoterpenes in needles and stem phloem, including β-phellandrene, δ-3-carene, γ-terpinene, and terpinolene. These trends were stronger in stem phloem than in needles. Lastly, chronic water stress inhibited the ability of *P. engelmannii* to form traumatic resin ducts in response to methyl jasmonate, a ubiquitous elicitor of inducible plant defenses. Collectively, our experiments show endogenous upregulation of SA in response to water stress in Engelmann spruce, resulting in changes to composition of volatile profiles and potentially suppressing induction of defense systems. These phenotypic changes may affect the performance of phloeophagous forest insects by altering plant resistance traits associated with defensive competency in response to herbivory.

**Data availability statement:** All data has been uploaded to the Dryad digital repository. The private-for-peer-review link can be found here: http://datadryad.org/share/LINK_NOT_FOR_PUBLICATION/ax6PH28-OLvEy2XtND-UnU9RDYiho7gXYAP-zYlQV31Y.

**Funding:** This research was funded by the National Science Foundation-Division of Integrative Organismal Systems (Award #2046109 to T.S.D.). The work of E.H. was supported by Award #1018490 from the National Institute of Food and Agriculture (United States Department of Agriculture). The funders had no role in study design, data collection and analysis, decision to publish, or preparation of the manuscript.

**Competing interests:** The authors declare no conflict of interest.

## Introduction

Environmental conditions place strong controls on plant physiology [1], which complicates the management of complex natural ecosystems such as forests. Although warming and increased atmospheric carbon concentrations can enhance plant productivity under some conditions [2], drought events are expected to increase in severity and frequency in the coming decades [3]. Multiple recent studies spanning temperate coniferous forest ecosystems in both North America and Europe confirm that drought events often precede outbreaks of phloem-feeding forest insects, especially bark beetles, which are important agents of biotic disturbance [4–6]. Analyses of geographic and climatic patterns indicates a strong role of both summer and winter droughts as triggers of subsequent bark beetle outbreaks [7]. Yet, whether or how plant physiological mechanisms drive these patterns is unclear and there is continued debate as to whether insect outbreaks following drought events occur because of changes in insect population demographics due to warmer conditions [8–10], or to changes in the ability of forest trees to defend themselves from herbivory due to drought [11,12], or some combination of these factors [4]. Accordingly, determining whether drought stress has subsequent effects on plant defense phenotypes may be important for evaluating why certain biological disturbances occur following drought.

In general, the defensive competency of conifer species in response to challenge by bark beetles and other forest insects are predicted by concentrations of secondary metabolites in different tissues [13,14], especially monoterpenes, as well as the ability to induce specific defensive traits in response to tissue damage [15–17]. For instance, the formation of resin ducts is often associated with conifer resistance to herbivory [18]. There are strong phylogenetic patterns underlying the distribution of resin ducts in Pinaceae: pines typically exhibit constitutive resin ducts and readily produce resin when phloem is damaged, but most spruces do not form resin ducts in phloem or sapwood until biotic challenge occurs [19,20]. The formation of inducible resin ducts is often signaled by the presence of specific hormones in phloem or needle tissues, especially jasmonic acid (JA) and salicylic acid (SA) [21]. Likewise, both hormones can cause changes in concentrations of secondary metabolites that play a role in resistance to herbivores and pathogens [22].

However, abiotic stressors such as drought also elicit hormonal responses in conifers that may interact with defense signaling [23], and in some cases the same hormone may mediate responses to both biotic and abiotic conditions. SA is a ubiquitous phytohormone and is upregulated in foliar tissues of many plants during drought stress, including conifers [24,25]. SA accumulation in foliar tissues can signal closure of stomatal apertures, limiting photosynthesis and gas exchange during periods of water stress [25,26]. Exogenous application of SA or the bioactive derivative methyl salicylate also elicits systemic acquired resistance by inducing production of phenols, condensed tannins [27] and terpenoid-rich resins [28], and SA signaling can be initiated by herbivore-associated molecular patterns [29]. Yet, it remains unknown whether drought-driven accumulation of multi-function hormones, such as SA, drive variation in conifer defense phenotypes.

In western North America, the (North American) spruce beetle, *Dendroctonus rufipennis* (Kirby) (Coleoptera: Curculionidae), is one of the most significant insect agents of forest tree mortality in alpine landscapes. Outside of boreal regions of Alaska and Canada, a primary host for *D. rufipennis* is Engelmann spruce (*Picea engelmannii* Parry ex Engelm.), which has a broad geographic distribution spanning the Canadian Rockies in western Canada to the Southern Rocky Mountains in the southwestern United States (Arizona and New Mexico). In the Southern Rocky Mountain region *D. rufipennis* outbreaks are especially severe, with millions of hectares of Engelmann spruce mortality reported by aerial surveyors in Colorado and Utah [30,31]. In this region in particular, drought has generally preceded outbreaks [5,7]. Previous work indicates that Engelmann spruce may be more susceptible to *D. rufipennis* than sympatric congeners (e.g., *P. pungens*, Colorado blue spruce) due to a relatively slow rate of monoterpene production in response to beetles and their symbiotic fungi [32]. However, it is presently unknown whether or how drought affects the chemical and physical defenses of Engelmann spruce, which could alter their ability to tolerate or resist herbivory by *D. rufipennis*.

To test the hypothesis that drought stress can alter defense phenotypes of Engelmann spruce, we employ a progressive series of experiments to test whether: (1) Engelmann spruce responds to water deprivation and water stress by increasing endogenous SA concentrations, (2) SA concentrations are associated with variation in monoterpene concentrations, and (3) whether drought stress interferes with the formation of traumatic resin ducts in response to methyl jasmonate application, a ubiquitous hormonal elicitor of conifer resistance traits. Our results indicate complex responses of Engelmann spruce secondary metabolites to drought-mediated SA accumulation, with consequences for defense phenotypes that could affect the behavior or performance of insect herbivores, including spruce beetle.

## Methods and materials

### Experiment 1: effect of acute water deprivation on water status, photosynthesis, and endogenous SA content

In our first experiment we tested the hypotheses that (1) acute water deprivation (i.e., water withholding) will lead to reduced stem water potential, (2) that stem water potential predicts photosynthetic rate, and (3) that stem water potential is associated with variation in salicylic acid content in different tissues. Three-year old Engelmann spruce seedlings were established in the greenhouse in 3.5 L pots and watered *ad libitum* for 3 months prior to the experimental period to ensure that seedlings were well-watered and vigorous. Only evidently healthy seedlings that showed evidence of new shoot growth in the early spring of current year were used (n = 24). To initiate the experiment, watering was stopped and the soil allowed to dry down until tree death occurred. Greenhouse conditions during the test were as follows: mean temperature range 17–23°C, 45% relative humidity, and a 16L:8D photoperiod. For this experiment and those described below, treatment positions were completely randomized on greenhouse benches.

Six replicate seedlings were randomly selected at each sample interval (0, 7, 21, and 28 d following water deprivation) for destructive sampling. At each sampling interval photosynthetic rate ($A_n$; μmol $CO_2$ fixed per m² s⁻¹) of replicates was measured from selected needles using an Li-6400 XT gas exchange analyzer (LiCor Bioscience, Lincoln, NE, USA). Next, seedlings were cut at the main stem and stem water potential ($\Psi_{stem}$) was measured using a Scholander pressure chamber (PMS Instruments, Albany, OR, USA), and tissues were immediately collected for analysis of SA profiles. Stem phloem (~2 g) and needles (~2 g) were excised using a scalpel and immediately flash-frozen in liquid N. Root balls were removed from pots and washed to remove excess soil, and a mix of fine and coarse roots were also collected (~5 g) and flash-frozen. Frozen tissues were kept at −20°C until analysis. After 28 d of water deprivation under greenhouse conditions all seedlings appeared dead and additional measurements of water potential could not be made, and no further samples were collected.

Tissue samples (roots, foliage, or stem phloem) were pulverized in liquid N, and 100 mg of powdered tissue was suspended in a 500 μl aliquot of methanol and acetic acid (99% MeOH, 1% glacial acetic acid, v/v). Suspensions were vortexed for 2 min, centrifuged for 1 min at 10,000 rpm, and then the supernatant was transferred to 1.5 ml Eppendorf

tubes by decanting. The pellet was re-extracted a second time and added to the first, for a total extract volume of ~1 ml. Resulting extracts were analyzed for SA as described below.

## Experiment 2: effects of drought stress on water status, SA content, and secondary metabolites

In a second experiment we tested the hypotheses that (1) chronic water stress results in differences in stem water potential; (2) differences in stem water potential predicts SA concentrations in stem phloem and needles; and (3) that concentration of SA predict concentration of secondary metabolites (monoterpenes) in respective tissues.

As above, Engelmann spruce seedlings were established in 3.5L pots and watered *ad libitum* for three months prior to the initiation of the experiment, as described above. At the start of the experiment, replicates were sorted randomly into two experimental watering groups, including an 'ample water' treatment (0.8 g water/g soil, n = 30) and a 'drought stress' group (0.2 g water/g soil, n = 30) after the methods of [33]. Experimental watering was applied every 72 h.

Trees in each water treatment group were randomly selected for destructive sampling at six intervals (0, 7, 14, 21, 28, and 35 days following initiation of watering treatments; n = 5 replicates per treatment group per day), at which time $\Psi_{stem}$ was measured and both stem phloem and foliar tissues were destructively sampled as described above. Methanol extracts were made of stem phloem and needles from each experimental unit, as above, to measure salicylic acid content. An additional hexane extract was also made to quantify monoterpenes. Approximately 100 mg of powdered tissue was suspended in a 500 µl aliquot of hexane (99% purity). Suspensions were vortexed for 2 min, centrifuged for 1 min at 10,000 rpm, and then the supernatant was transferred to 1.5 ml Eppendorf tubes by decanting. The pellet was re-extracted a second time and added to the first, for a total extract volume of ~1 ml. Resulting hexane extracts were analyzed, respectively, for target monoterpenes including (+)-α-pinene, (-)-β-pinene, (+)-3-carene, camphene, β-phellandrene, sabinene, (+)-linalool, β-myrcene, γ-terpinene, and terpinolene. Target analytes were selected based on earlier studies characterizing monoterpene composition of mature *P. engelmannii* under field conditions [32,34]. Stem water potentials were also recorded at the time of destructive sampling for each replicate.

## Experiment 3: effects of drought stress on formation of traumatic resin ducts

Experimental conditions were identical to those described above, with 'drought stress' and 'ample water' (i.e., the control treatment) (0.2 and 0.8 g $H_2O$/g soil, respectively) treatments applied via gravimetric watering. However, a defense elicitor (methyl jasmonate, MeJa) was applied to all replicates to stimulate formation of resin ducts. A variety of earlier studies have demonstrated that exogenous MeJa application catalyzes the production of inducible defenses in conifers via formation of 'traumatic resin ducts' in stem phloem and xylem [21,34–37], and numerous others]. Accordingly, MeJa is widely used in studies on induced defenses in conifers and is well-known to induce resin duct formation. Prior to the present experiment we confirmed that MeJa application stimulates resin duct formation in mature *P. engelmannii* in the field by spraying trees and monitoring resin duct production over time [38].

At the onset of the experiment, all replicate seedlings were treated once with 2 ml of a 100 mM solution of methyl jasmonate (MeJa) amended with 0.1% Tween 20, delivered to the foliage and stem via a spray bottle. Spray delivery was calibrated prior to application. Replicates were then randomly selected for destructive sampling at five intervals post-MeJa application (i.e., 7, 14, 21, 28, and 35-d post application). Replicates selected for sampling at each interval were cut at a height of 3 cm above the soil line, and a stem section 5 cm in length was excised. A microtome was used to cut a thin cross-sectional slice of each excised stem section to ~9 µm thickness for anatomical analysis. Sliced stem sections were mounted on a glass slide with water (100 µl) and a coverslip and were immediately photographed under a slide microscope at 400x magnification. Slide images were subsequently categorized based on the criteria described in [32] to assign resin duct formation. Briefly, stem sections were scored as having no resin ducts present in the most recent growth increment (Class 0), having tangential alignment of 2 or more resin ducts (Class I), tangential alignment of resin ducts bisecting 20–90% of the increment (Class II), or tangential alignment of resin ducts that bisected >90% of the increment (Class III).

                                                

## Quantification of salicylic acid and monoterpenes in tissue extracts

Samples were assigned random labels so that laboratory analyses were conducted blindly, preventing potential bias. Methanol extracts were transferred to a 96 well plate (Waters) and sealed with clear film covers to prevent evaporation. Study pooled quality control (SPQC) samples were created by mixing equal volume aliquots from each unknown sample in each sample set. Various calibrators were used as quality control samples. All calibration standards (C1-C8) were prepared in methanol.

Ultra-performance liquid chromatography (UPLC) separation of calibrators and experimental samples was performed on a Waters Acquity H-Class UPLC instrument equipped with a quaternary pump and an Acquity UPLC® BEH C18 column (2.1 x 100 mm, 1.7µm particle size, part 186002352) including a Van Guard™ UPLC BEH C18 pre-column (2.1 × 5 mm, 1.7 µm particle size, part 186003975). Column temperature was 35.0°C, while the autosampler was kept at 10.0°C; injections were 4 µL, and mobile phase flow rate was 0.4 ml/min. The UPLC was in line with a Waters Xevo TQD (triple quadrupole) Zspray ESI (electrospray ionization) mass spectrometer, which was controlled by MassLynx software (version 4.2). The mass spectrometer was operated in ESI-negative mode, at a capillary voltage of 1.50 kV, with a desolvation temperature of 650°C, a desolvation gas flow of 1000 L/h, a cone source gas flow of 0 L/h, and a source temperature of 150°C. Gradient separation was performed over 7 minutes using the following solvent channels: A: 10mM ammonium acetate in water, B: 10mM ammonium acetate in acetonitrile (Table S1 in S1 File). The column was equilibrated for at least 4 minutes in the starting conditions before use. Calibration standards, calibrator QCs, pooled QCs, and methanol solvent blanks were acquired at the start, middle and end of each plate acquired.

Multiple reaction monitoring was used to quantify hormone content in the extracts. A combination of literature and empirically determined transition ions were used for optimal quantitation [39–40]. The quantifier ion (first transition listed) and qualifier ion (second transition listed) for SA, retention time, cone voltage, and collision energy are given in Table S2 in S1 File. Dwell times were 0.034 s.

All monoterpene standards were purchased from Fischer-Scientific (Hampton, NH, USA). Hexane extracts were transferred to 2 ml GC/MS vials (Agilent) and stored at 4°C to prevent evaporation. Study pooled quality control (SPQC) samples were created by mixing equal volume aliquots from each unknown sample in each sample set. Various calibrators were used as quality control samples. All calibration standards were prepared in hexane.

Gas chromatography (GC) separation of calibrators and experimental samples was performed on a ThermoScientific TRACE 1310 gas chromatograph equipped with a Zebron™ ZB-5HT Inferno™ column (30 m length, 0.25 mm I.D., 0.25 µm film thickness, part 7HG-G015-11). The GC was in line with a ThermoScientific ISQ-QD mass spectrometer using chemical ionization (CI), which was controlled by Chromeleon software version 7. The injections (1 µl) onto the column were automated, and the GC/MS was operated in splitless mode with a front inlet temperature of 300°C, carrier gas was helium at a flow rate of 1.2 mL/min. Gradient separation was performed over 34.667 minutes, and the temperature program was as follows: 40°C for 5 min and increasing by 10°C per min thereafter to 250°C with a 5 min hold time, and then increasing by 30°C per min until 300°C with a final 2 min hold time (Table S3 in S1 File). The oven was equilibrated 0.10 minutes in the starting conditions before use. Calibration standards, calibrator QCs, pooled QCs, and hexanes solvent blanks were acquired at the start, middle, and end of each sample set. Monoterpenes in extracts were confirmed by comparing electron mass spectra and retention times against authentic standards and NIST libraries. Empirically determined transition ions were used for optimal quantitation. The quantifier ion (first transition listed) and qualifier ions (subsequent transitions listed) for each terpene, as well as their retention times, are given in Table S4 in S1 File.

Following acquisition, data files were imported into Skyline Targeted Mass Spec Environment for processing [41]. Retention time matching was manually inspected, and a bilinear regression weighted $1/x^2$ calibration was created for SA (0.05 to 100 µg/mL or 0.06−120 µg/mL global range; see Table S5 in S1 File for calibration range) and each terpene standard (0.005–0.1 mg/mL global range). Accordingly, detection limits ranged from approximately 15–20 ppb for salicylic

acid and 1.5–2 ppm for terpene compounds, with quantification limits of 50–60 ppb and 5 ppm, respectively. The SA and monoterpene concentration of each sample was calculated relative to the peak area for external standards. When data processing in Skyline was completed, data were further checked, prior to statistical analysis, to determine if the acquired data for each plate met pre-set quality control criteria. The following criteria were required to meet QC: $R^2$ value above 0.95 with a minimum of 6 points in the calibration curve, and 75% of the non-zero calibrators including the LOQ within ± 20% of theoretical concentrations.

## Statistical analysis

All statistical analyses were performed using the JMP 17.0 software (SAS Institute, Cary, NC) and a type I error rate of $\alpha = 0.10$ for assigning statistical significance to modeled effects [42–43]. Prior to implementing parametric analyses, assumptions of normality and homoscedasticity were evaluated.

In Experiment 1, variation in stem water potential was analyzed due to the effect of time (day of experiment) using one-way ANOVA to test whether withholding water affects stem water potential. Simple linear models were used to test the hypothesis that variation in stem water potential predicts photosynthetic rate (µmol CO2 fixed per m² s⁻¹), and variation in SA concentrations (µg/g) in roots, needles, and stem phloem. One-way ANOVA was also used to test the hypothesis that mean SA concentrations vary between tissue types (roots, needles, and stem phloem).

In Experiment 2 we first tested whether seedlings under chronic water stress (0.2 g $H_2O$/g soil) express lower mean stem water potential than seedlings that are well-watered (0.8 g $H_2O$/g soil) using a two-way ANOVA model analyzing time (day-of-test) and the watering treatment nested within day as fixed effects on the response of stem water potentials. As in the first experiment, we used simple linear regression to test the hypothesis that water potentials predict variation in SA concentration in both needles and stem phloem. We also used correlation analysis (Pearson's $r$) to test whether there was a relationship between SA concentrations in needles and stem phloem. Next, we used two-way ANOVA to test whether SA concentrations interact with watering treatments to predict variation in concentrations of individual and pooled (summed total) monoterpenes in both needles and stem phloem. The sample distribution of monoterpenes that represented a relatively small fraction of the overall profile tended to be skewed left, especially in foliar tissues. However, ANOVA procedures generally remain robust to type I errors with large sample sizes [44]; here, n=60 and 59 for foliar and stem phloem, respectively. Nonetheless, sample standard deviations for monoterpene concentrations were comparable among watering groups.

In Experiment 3 a nominal logistic model was used to analyze variation in the proportion of seedlings in each watering treatment group (0.2 or 0.8 g $H_2O$/g soil) forming Type I, II, and III resin ducts following exposure to the elicitor MeJa, treating time (weeks following treatment) and treatment nested within week as fixed effects.

## Results

### Experiment 1: effect of acute water deprivation on water status, photosynthesis, and endogenous SA content

Water deprivation strongly affected $\Psi_{stem}$; water potential decreased in stems over time following water withholding ($F_{3,21} = 7.142$, $P = 0.001$; $R^2 = 0.505$; Fig 1A). Photosynthetic rate was strongly predicted by variation in $\Psi_{stem}$, and photosynthesis declined sharply as $\Psi_{stem}$ fell below −2.0 MPa ($F_{1,23} = 10.429$, $P = 0.003$; $R^2 = 0.311$; Fig 1B). SA concentration varied strongly between needle, stem phloem, and root tissues ($F_{2,77} = 37.430$; $P < 0.001$; $R^2 = 0.492$). Needles had the highest concentrations of SA (mean = 0.845 µg/g foliar biomass); concentrations were intermediate in stem phloem (mean = 0.344 µg/g stem phloem biomass) and lowest in root tissues (mean = 0.017 µg/g root biomass; Fig 1C). In needle tissues, SA concentrations increased as $\Psi_{stem}$ declined ($F_{1, 24} = 4.440$, $P = 0.046$; $R^2 = 0.168$, Fig 1D), but there was not a clear relationship between $\Psi_{stem}$ and SA concentration in roots ($F_{1, 24} = 1.961$, $P = 0.171$; $R^2 = 0.063$) nor stem phloem ($F_{1, 24} = 1.808$, $P = 0.191$; $R^2 = 0.072$).

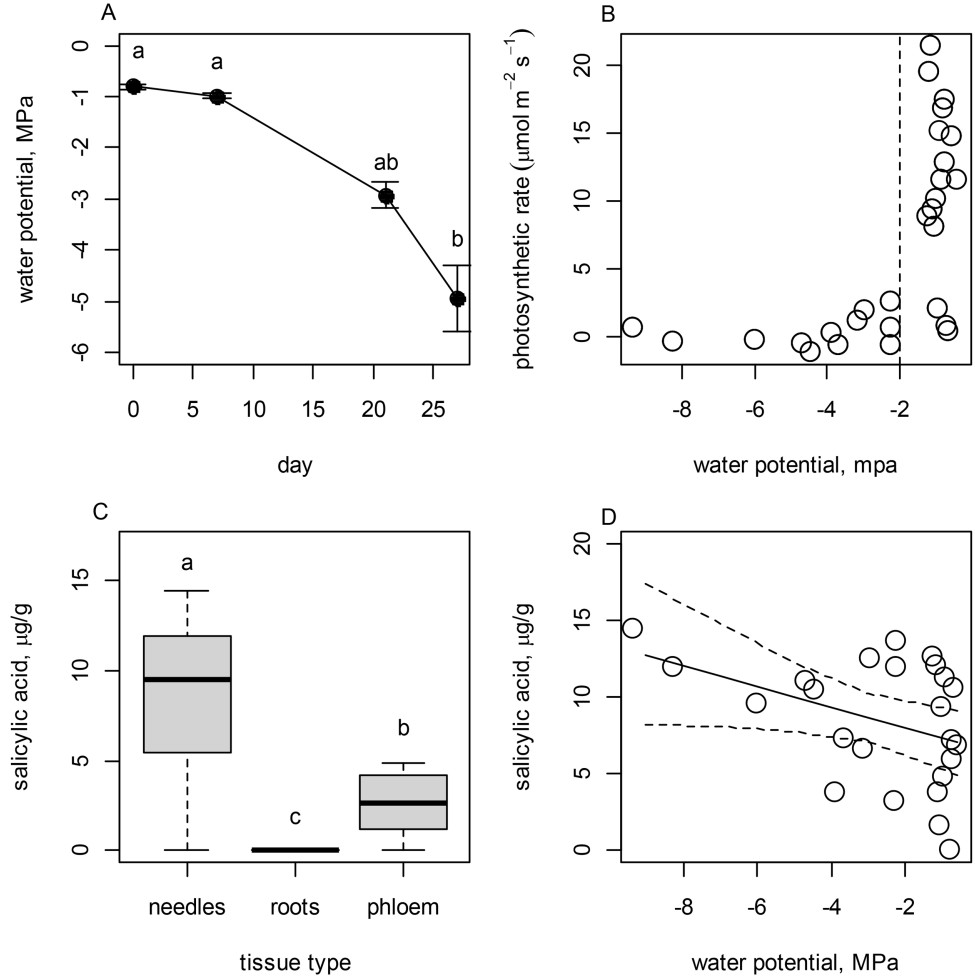

**Fig 1. The effects of water deprivation on Engelmann spruce.** (A) Change over time in stem water potential of *Picea engelmannii* seedlings during a water deprivation experiment; (B) the relationship between stem water potential and photosynthetic rate (µmol m$^{-2}$s$^{-1}$); (C) the distribution of salicylic acid (SA) concentrations in needles, roots, and stem phloem; and (D) the relationship between stem water potential and SA concentrations in needle tissues. The solid line shows a fitted regression model and dashed lines represent the 95% CI. Where lowercase letters are shown, they denote significant differences among means identified by Tukey's HSD test. Water deprivation caused strongly negative water potentials, which corresponded to reduced photosynthesis and elevated SA in needles.

## Experiment 2: effects of drought stress on water status, SA content, and secondary metabolites

Watering treatment nested by day significantly affected variation in mean $\Psi_{stem}$ ($F_{6, 47} = 12.490$; $P < 0.001$), but there was no evidence that time (day of test) alone affected mean $\Psi_{stem}$ ($F_{5,47} = 1.657$; $P = 0.163$). Mean $\Psi_{stem}$ remained relatively constant over time in seedlings receiving ample water but mean $\Psi_{stem}$ gradually declined near the threshold for loss of photosynthetic function, in seedlings receiving the drought treatment (Fig 2A), confirming that the drought stress treatment resulted in reduced mean water potentials. There was evidence that SA concentrations in needles were predicted by variation in $\Psi_{stem}$ ($F_{1, 58} = 4.062$; $P = 0.048$; $R^2 = 0.065$), however, concentrations in stem phloem were not related to $\Psi_{stem}$ ($F_{1, 58} = 0.120$; $P = 0.730$; $R^2 = 0.002$; Fig 2B). There was no evidence of a correlation between SA concentrations in needles and SA concentrations in stem phloem (Pearson's $r = 0.160$; $P = 0.229$, Fig 2C).

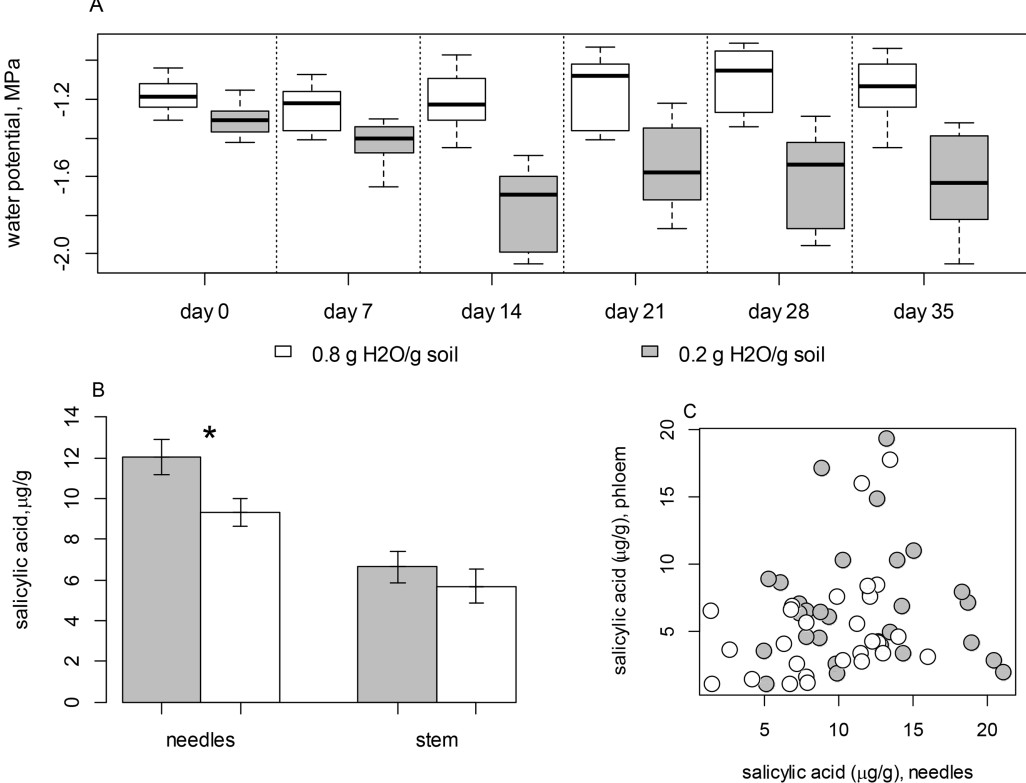

**Fig 2. Relationship between drought stress and salicylic acid (SA) levels in Engelmann spruce tissues.** (A) Variation in stem water potentials (MPa) over time in *Picea engelmannii* seedlings receiving ample (0.8 g $H_2O$/g soil, white symbols) and drought (0.2 g $H_2O$/g soil, gray symbols) watering treatments. (B) Mean concentrations of SA in needle and stem tissues of *P. engelmannii* seedlings relative to watering treatments. The asterisk denotes a significant difference in SA concentrations between drought-stressed and well-watered seedlings. (C) The relationship between SA concentrations in needles and stem phloem. As was the case for acute water deprivation, chronic drought stress results in elevated SA concentrations in needles.

In needle tissues, mean concentrations of most monoterpenes were unresponsive to variation in SA or watering treatments. A SA concentration × watering treatment interaction significantly affected variation in β-phellandrene concentrations, which increased with the concentration of SA, but only in drought-stressed seedlings ($P = 0.017$; Fig 3A). In addition, terpinolene concentrations increased with the concentrations of SA regardless of watering treatments ($P = 0.039$; Fig 3B).

In stem phloem tissues, monoterpenes were responsive to a SA concentration × watering treatment interaction, including 3-carene ($P = 0.058$; Fig 4A), γ-terpinene ($P = 0.091$; Fig 4B), sabinene ($P = 0.077$; Fig 4C), and terpinolene ($P = 0.055$; Fig 4D). In each instance, monoterpene concentrations increased with SA concentrations in stem phloem, but only for drought-stressed seedlings. Other stem phloem monoterpenes, and total monoterpenes, were not otherwise responsive to variation in SA concentrations, watering treatments, or their interaction (Table 1).

### Experiment 3: effects of drought stress on formation of traumatic resin ducts

Time following application of methyl jasmonate had a significant effect on the proportion of the seedlings forming traumatic resin ducts, with a greater proportion of Type II and III resin ducts forming over time ($\chi^2 = 90.028$; df = 12; log worth=13.312; $P < 0.001$). In addition, there was a significant effect of watering treatment nested within week ($\chi^2 = 23.823$; df = 15; log worth=1.167; $P = 0.068$), with a higher frequency of Type III ducts forming in seedlings receiving the ample water (0.8 g $H_2O$/g soil) treatment (Fig 5).

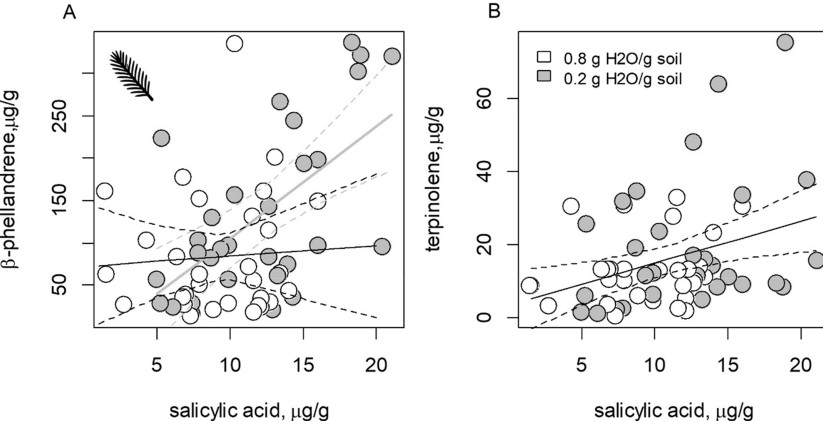

**Fig 3. Drought stress and salicylic acid (SA) interact to affect expression of foliar monoterpenes.** The effects of (A) SA × watering treatment interaction on β-phellandrene concentrations and (B) SA alone on terpinolene concentrations. In each panel dashed lines show the 95% CI of each fit. In panel A, the gray line shows the fitted model for seedlings receiving the drought-stress treatment and the black line shows the fitted model for well-watered seedlings; in panel B, only SA was a significant predictor of secondary metabolite concentrations.

## Discussion

Here we linked the changes in water status of conifer tissues to both photosynthetic rate and SA concentrations and subsequently show that variation in stem water potential ($\Psi_{stem}$) and SA regulate expression of terpene profiles and formation of induced defense structures in response to hormonal elicitors. Experimentally applied water stress, both chronic and acute, reduced $\Psi_{stem}$ of Engelmann spruce seedlings (Fig 1A), and values below −2.0 MPa were associated with a substantial reduction in net photosynthesis ($A_n$, Fig 1B) and elevated concentrations of SA, especially in needle tissues (Fig 1C). However, SA concentration in root tissues did not vary with stem water potential [24], lending support to the hypothesis that accumulation of SA occurs primarily in needles and acts to induce stomatal closure and ameliorate drought stress, potentially by reducing transpiration [26,45]. Indeed, as physiological water stress intensified, SA concentrations in needles increased linearly (Fig 1D). This corresponds well to earlier studies (e.g., [24]) demonstrating a protective effect of SA on plants under drought stress. For example, water content of Chinese nutmeg (*Torreya grandis*, Taxaceae) exposed to moderate drought was higher in individuals treated exogenously with SA than in control individuals, though this pattern was not observed in severely stressed individuals [46].

The differences in SA concentrations between drought-stressed and well-watered replicates were more pronounced in needles than in stem phloem (Fig 2B), and there was no clear correlation between SA concentrations in stems and needles (Fig 2C). This could indicate that different environmental factors drive accumulation of SA in needle and stem tissues [47]. Nonetheless, SA content interacted with watering treatments in both needles and stem phloem to affect variation in specific monoterpenes (β-phellandrene, δ-3-carene, γ-terpinene, sabinene and terpinolene), and these trends were generally stronger in phloem than in needles (Figs 3 and 4).

Similar induction of specific monoterpenes in response to SA accumulation is reported in other studies (e.g., [48]). Monoterpenes varied in response to SA only in drought-stressed replicates, and β-phellandrene and the isomers γ-terpinene and terpinolene (sometimes referred to as δ-terpinene) are very structurally similar, and all are formed from the precursors dimethylallyl diphosphate (DMADP) and isopentenyl pyrophosphate (IPP). The genes encoding DMADP and IPP are often overexpressed during periods of environmental stress [49] and during biotic challenges [50] Yet, our data show that total monoterpene concentrations were comparable between drought-stressed and well-watered replicates, indicating drought modulated monoterpene composition more so than total concentrations. This matches previous

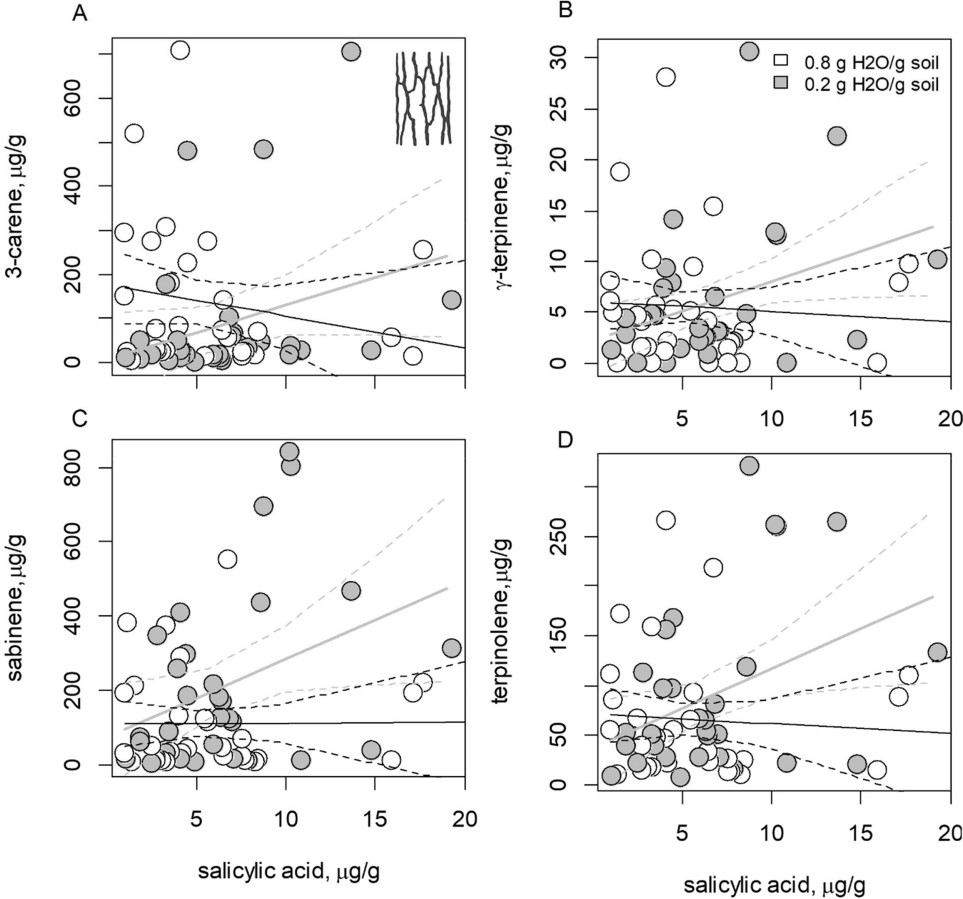

**Fig 4. Drought stress and salicylic acid (SA) interact to affect expression of phloem monoterpenes.** The effects of SA × watering treatment interaction on the concentrations of secondary metabolites in the stem phloem of *Picea engelmannii* seedlings, including (A) 3-carene, (B) γ-terpinene, (C) sabinene, and (D) terpinolene. In each panel dashed lines show the 95% CI of each fit. Gray lines show fitted models for the drought-stress treatment and black lines show fitted models for well-watered seedlings. In drought-stressed seedlings, SA is positively correlated with concentration of specific stem phloem monoterpenes.

studies demonstrating that monoterpene biosynthesis in plants is SA independent, although hydroxylated monoterpenes can induce SA accumulation [51]. SA accumulation can trigger synthesis of specific monoterpenes in a few plant species [52], and it is possible that upregulation of certain monoterpenes along with SA in *P. engelmannii* needles results from shared target molecules (e.g., bacterial pathogens; [48]). SA accumulation leads to stomatal closure through induction of ROS species in tissue [25–26], and monoterpenes act as scavengers of reactive oxygen species (ROS), increasing thermotolerance of plants following cessation of transpiration [53]. Hence, we hypothesize that the observed interaction between SA and monoterpenes acts to reduce ROS damage to chloroplasts while potentially augmenting drought tolerance through stomatal closure.

Our experiments were performed in a greenhouse where thermal conditions were relatively constant during the experimental period, but in field studies monoterpene concentrations (and emissions) typically fluctuate during the growing season and are elevated under higher temperatures [47,54,55]. Other studies on spruce chemical defenses suggest that drought can both elevate or reduce terpene concentrations in *P. abies* needles, but these effects are strongly genotype-dependent (e.g., [56]). Studies of monoterpene toxicity to forest insects such as bark beetles demonstrate

**Table 1. Summary of two-way ANOVA models.**

| Tissue | Monoterpene | Source | df | SS | *F* | *P* |
|---|---|---|---|---|---|---|
| Needles | α-pinene | SA concentration | 1 | 0.047 | 0.326 | 0.569 |
| | | Watering treatment | 1 | 0.142 | 0.967 | 0.329 |
| | | SA x watering | 1 | 0.178 | 1.216 | 0.274 |
| | | error | 56 | 8.228 | – | – |
| | β-phellandrene | **SA concentration** | **1** | **2.263** | **8.670** | **0.004** |
| | | Watering treatment | 1 | 0.525 | 2.011 | 0.161 |
| | | **SA x watering** | **1** | **1.575** | **6.035** | **0.017** |
| | | error | 56 | 0.261 | – | – |
| | β-pinene | SA concentration | 1 | 0.000 | 0.102 | 0.750 |
| | | Watering treatment | 1 | 0.001 | 0.238 | 0.627 |
| | | SA x watering | 1 | 0.000 | 0.178 | 0.674 |
| | | error | 56 | 0.283 | – | – |
| | camphene | SA concentration | 1 | 0.663 | 0.806 | 0.372 |
| | | Watering treatment | 1 | 0.629 | 0.764 | 0.385 |
| | | SA x watering | 1 | 0.852 | 1.036 | 0.313 |
| | | error | 56 | 46.050 | – | – |
| | 3-carene | SA concentration | 1 | 0.000 | 0.071 | 0.790 |
| | | Watering treatment | 1 | 0.000 | 0.040 | 0.841 |
| | | SA x watering | 1 | 0.006 | 1.380 | 0.245 |
| | | error | 56 | 0.270 | – | – |
| | γ-terpinene | SA concentration | 1 | 0.004 | 0.232 | 0.631 |
| | | Watering treatment | 1 | 0.016 | 0.903 | 0.345 |
| | | SA x watering | 1 | 0.002 | 0.114 | 0.736 |
| | | error | 56 | 1.025 | – | – |
| | linalool | SA concentration | 1 | 0.036 | 0.061 | 0.805 |
| | | Watering treatment | 1 | 0.131 | 0.218 | 0.641 |
| | | SA x watering | 1 | 1.235 | 2.060 | 0.156 |
| | | error | 56 | 33.582 | – | – |
| | myrcene | SA concentration | 1 | 0.309 | 0.228 | 0.634 |
| | | Watering treatment | 1 | 2.589 | 1.907 | 0.172 |
| | | SA x watering | 1 | 0.029 | 0.021 | 0.884 |
| | | error | 56 | 76.022 | – | – |
| | sabinene | SA concentration | 1 | 0.000 | 0.201 | 0.655 |
| | | Watering treatment | 1 | 0.001 | 0.623 | 0.433 |
| | | SA x watering | 1 | 0.001 | 0.845 | 0.361 |
| | | error | 56 | 0.113 | – | – |
| | terpinolene | **SA concentration** | **1** | **0.219** | **4.462** | **0.039** |
| | | Watering treatment | 1 | 0.039 | 0.808 | 0.372 |
| | | SA x watering | 1 | 0.050 | 1.037 | 0.312 |
| | | error | 56 | 2.749 | – | – |
| | Total monoterpenes | SA concentration | 1 | 0.546 | 1.125 | 0.293 |
| | | Watering treatment | 1 | 0.000 | 0.000 | 0.985 |
| | | SA x watering | 1 | 0.349 | 0.720 | 0.399 |
| | | error | 56 | 27.179 | – | – |
| Stem | α-pinene | SA concentration | 1 | 1.921 | 1.863 | 0.177 |

*(Continued)*

**Table 1.** (Continued)

| Tissue | Monoterpene | Source | df | SS | F | P |
|---|---|---|---|---|---|---|
| phloem | | Watering treatment | 1 | 2.023 | 1.963 | 0.166 |
| | | SA x watering | 1 | 1.355 | 1.314 | 0.256 |
| | | error | 56 | 56.699 | – | – |
| | β-phellandrene | SA concentration | 1 | 0.005 | 0.006 | 0.937 |
| | | Watering treatment | 1 | 0.318 | 0.396 | 0.531 |
| | | SA x watering | 1 | 0.198 | 0.246 | 0.621 |
| | | error | 56 | 44.258 | – | – |
| | β-pinene | SA concentration | 1 | 0.698 | 1.122 | 0.293 |
| | | Watering treatment | 1 | 0.569 | 0.916 | 0.342 |
| | | SA x watering | 1 | 0.088 | 0.142 | 0.707 |
| | | error | 56 | 34.190 | – | – |
| | camphene | SA concentration | 1 | 0.014 | 1.697 | 0.198 |
| | | Watering treatment | 1 | 0.002 | 0.315 | 0.576 |
| | | SA x watering | 1 | 0.005 | 0.666 | 0.417 |
| | | error | 56 | 0.485 | – | – |
| | 3-carene | SA concentration | 1 | 0.421 | 0.258 | 0.613 |
| | | Watering treatment | 1 | 2.205 | 1.352 | 0.249 |
| | | **SA x watering** | **1** | **6.064** | **3.719** | **0.058** |
| | | error | 56 | 89.677 | – | – |
| | γ-terpinene | SA concentration | 1 | 2.284 | 1.469 | 0.226 |
| | | Watering treatment | 1 | 0.075 | 0.049 | 0.825 |
| | | **SA x watering** | **1** | **4.491** | **2.943** | **0.091** |
| | | error | 56 | 83.922 | – | – |
| | linalool | SA concentration | 1 | 0.181 | 0.123 | 0.726 |
| | | Watering treatment | 1 | 0.795 | 0.541 | 0.465 |
| | | SA x watering | 1 | 0.148 | 0.100 | 0.752 |
| | | error | 56 | 80.880 | – | – |
| | myrcene | SA concentration | 1 | 0.118 | 0.177 | 0.675 |
| | | Watering treatment | 1 | 0.043 | 0.064 | 0.800 |
| | | SA x watering | 1 | 0.056 | 0.084 | 0.772 |
| | | error | 56 | 36.714 | – | – |
| | sabinene | **SA concentration** | **1** | **4.588** | **3.333** | **0.073** |
| | | **Watering treatment** | **1** | **4.822** | **3.503** | **0.066** |
| | | **SA x watering** | **1** | **4.444** | **3.228** | **0.077** |
| | | error | 56 | 75.718 | – | – |
| | terpinolene | SA concentration | 1 | 3.227 | 2.352 | 0.130 |
| | | Watering treatment | 1 | 1.633 | 1.190 | 0.279 |
| | | **SA x watering** | **1** | **5.259** | **3.833** | **0.055** |
| | | error | 56 | 75.457 | – | – |
| | Total monoterpenes | SA concentration | 1 | 0.016 | 0.018 | 0.893 |
| | | Watering treatment | 1 | 1.328 | 1.443 | 0.234 |
| | | SA x watering | 1 | 2.167 | 2.355 | 0.130 |
| | | error | 56 | 50.630 | – | – |

Summary of models analyzing variation in individual and total monoterpene concentrations of needles and stem phloem *in Picea engelmannii* seedlings relative to salicylic acid (SA) concentrations, watering treatments (0.2 and 0.8 g H2O/g soil), and their interaction. Significant effects (*P*<0.10) denoted in bold text.

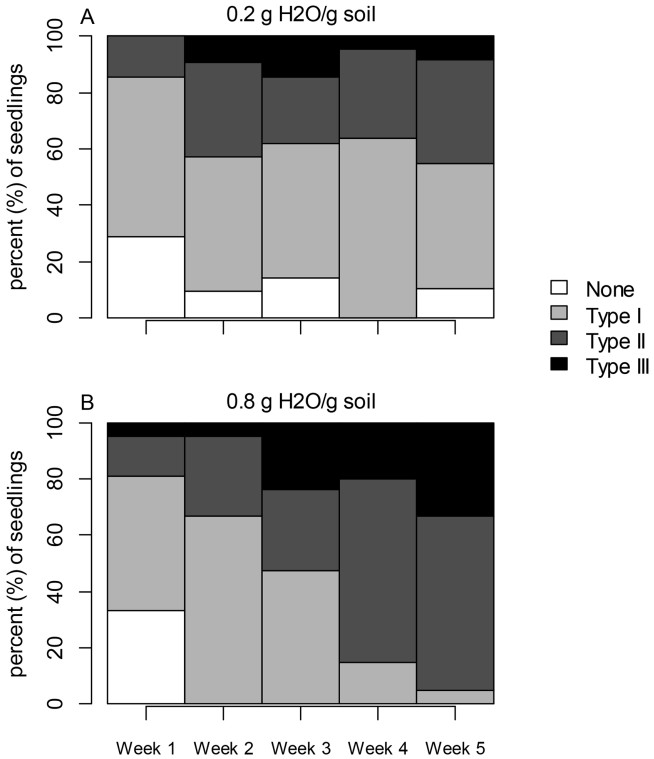

**Fig 5. Drought stress limits the ability of Engelmann spruce to form resin ducts.** The frequency of traumatic resin ducts formation in *Picea engelmannii* seedlings over time following exposure to the elicitor methyl jasmonate (MeJa) for (A) droughted and (B) well-watered seedlings. In well-watered seedlings a higher proportion of individuals formed traumatic resin ducts.

that specific compounds, or blends of compounds, can have differing effects on insect survival [57–59] as well as the performance of their symbiotic microorganisms [34]. Therefore, the drought-driven changes in secondary metabolite composition we report here could have consequences for the function of resin-based defenses, though this has not been extensively tested with *D. rufipennis*, the primary tree-killing bark beetle associated with *P. engelmannii*.

When MeJa was applied as an elicitor of traumatic resin duct formation [21,38,60,61], there was clear evidence that tangentially aligned resin ducts were formed rapidly in sapwood following application. However, duct formation was less extensive in drought-stressed replicates (Fig 5), indicating that water limitation interacts with the ability to respond to hormones signaling production of inducible defenses. Several potentially competing hypotheses may explain this pattern. First, our results showed that drought stress was associated with elevated SA, but also a reduction in formation of traumatic resin ducts (Figs 1, 2 and 5). It is possible that SA inhibits the ability of *P. engelmannii* to respond to MeJA or other hormonal elicitors of defense consistent with a pattern of 'hormone crosstalk' demonstrated in other plant systems [62–63] whereby upregulation of certain phytohormones negatively affects the ability to respond to other hormonal signals. Cellular mechanisms underlying such hormone antagonism are poorly described but could occur due to shared receptors or a loss of sensitivity in response to elicitors over time [64]. However, drought-driven stomatal closure can also lead to carbon starvation in plants, halting primary and secondary metabolism(McDowell 2011). In general, formation of both constitutive and inducible resin ducts is metabolically costly and an important predictor of conifer survival of bark beetle attack [18,65,66]. Yet, an inability to photosynthesize during drought stress may constrain available metabolic resources required for resin duct formation. Our results suggest that drought-stressed *P. engelmannii* may be less resistant to insect pressure

due to an inability to form extensive resin duct systems either as a result of antagonistic hormone crosstalk, carbon starvation, or both factors simultaneously, but this cannot be determined from our experimental design. Further studies are merited to disentangle these potential mechanistic effects.

Accordingly, environmental pressures that alter the ability of conifers to respond to certain hormones could partially explain why outbreaks of *D. rufipennis* are reported in *P. engelmannii* after regional droughts [5]. However, drought-associated changes in secondary metabolite blends could also affect the apparency (i.e., the relative likelihood that suitable host plants are found by herbivores) of trees to beetles, making them easier to identify and locate during the host-searching phase [67–68]. Our data show that certain monoterpenes were related to drought stress, which could provide a reliable evolutionary signal for identifying susceptible or drought-stressed hosts, and numerous previous studies have demonstrated that specific monoterpene blends drive beetle responses to host volatiles [69–72]. Therefore, it will be critical to differentiate whether the monoterpenes we report as 'responsive' to SA in drought-stress trees have bearing for the defensive competency of *P. engelmannii*, whether they synergize attraction to other host kairomones or aggregation pheromones, or whether both patterns can occur. Earlier studies report relatively low discrimination of *D. rufipennis* to variation in kairomonal blends in the presence of pheromonal attractants [73], suggesting that drought-related volatile shifts in *P. engelmannii* may only be important for 'pioneer' beetles that initiate mass attacks. Subsequent studies also report that ethanol, which accumulates in and elutes from tissues of drought-stressed conifers [74–75], can strongly synergize *D. rufipennis* attraction to pheromones and kairomone blends [76].

In summary, our experiments show that both acute and chronic drought stress reduces stem water potentials in *P. engelmannii* over time, corresponding to increased SA content in needles more so than in phloem. SA content in both needles and phloem interacted with water status to predict variation in specific monoterpenes, and drought stress inhibited the ability of trees to form traumatic resin ducts. These collective changes in the defensive phenotypes of forest trees could have importance for mediating plant-insect interactions with tree-killing bark beetles through various means, including shifts in both host attractiveness and defensive competency. An assessment of whether similar responses occur in the field presents interpretive challenges as monoterpene concentrations and blends may vary within individual trees [77], and relatively few studies have focused on these compounds in *P. engelmannii* phloem [32,34,78]. Nonetheless, such field studies are needed for a more complete understanding of how environmental stressors such as drought affect the defenses of long-lived mature conifers, which must contend with repeated biotic challenge. More broadly our results present an example by which stress may mediate key plant traits, potentially altering interactions with herbivores via cascading effects of hormonal responses. Under current projections of a warmer and drier climate [79–80], such patterns may have widespread relevance for trends of plant mortality across landscapes as well as the development and selection of varietals in agriculture and forestry.

## Supporting information

**S1 File. Table S1: Gradients used in chromatographic separation (UPLC); Table S2: Ion transitions used in quantification of salicylic acid; Table S3: Gradients used in chromatographic separation of monoterpenes (GC); Table S4: Ion transitions used in quantification of monoterpenes; Table S5: Dilutions of standards used in quantification of salicylic acid.**
(DOCX)

## Acknowledgments

We are grateful to Nathaniel Comai for assistance with laboratory and greenhouse work and sample processing for resin duct determination. Drs. Claudia Boot and Karolien Denef (Colorado State University) oversaw data acquisition of hormone and terpene data in the Analytical Resource Core facility at Colorado State University.

## Author contributions

**Conceptualization:** Thomas Seth Davis, Troy Ocheltree.

**Data curation:** Thomas Seth Davis, Ehsan Khedive.

**Formal analysis:** Thomas Seth Davis.

**Funding acquisition:** Thomas Seth Davis.

**Investigation:** Thomas Seth Davis, Edward Hill, Troy Ocheltree.

**Methodology:** Thomas Seth Davis, Edward Hill.

**Project administration:** Thomas Seth Davis.

**Visualization:** Thomas Seth Davis.

**Writing – original draft:** Thomas Seth Davis.

**Writing – review & editing:** Ehsan Khedive, Edward Hill, Troy Ocheltree.

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
