## [Decision Letter · Decision Letter 0]

7 Dec 2025

PONE-D-25-50354Water stress induces salicylic acid accumulation, altering monoterpene profile and suppressing resin duct formation in Engelmann sprucePLOS One

Dear Dr. Davis,

Thank you for submitting your manuscript to PLOS ONE. After careful consideration, we feel that it has merit but does not fully meet PLOS ONE’s publication criteria as it currently stands. Therefore, we invite you to submit a revised version of the manuscript that addresses the points raised during the review process.

We look forward to receiving your revised manuscript.

Kind regards,

Mansureh Ghavam

Academic Editor

PLOS One

**Journal Requirements:**

3. . Thank you for stating the following financial disclosure:

“This research was funded by the National Science Foundation-Division of Integrative Organismal Systems (Award #2046109 to T.S.D.). The work of E.H. was supported by Award #1018490 from the National Institute of Food and Agriculture (United States Department of Agriculture).”

4. Please note that funding information should not appear in any section or other areas of your manuscript. We will only publish funding information present in the Funding Statement section of the online submission form. Please remove any funding-related text from the manuscript.

6. We note that you have referenced “Davis TS, unpublished data; Figure S1 and Lerdau et al. 1997, Michelozzi et al. 2008, Vanhatalo et al. 2015, Khedive et al, unpublished data” which has currently not yet been accepted for publication. Please remove this from your References and amend this to state in the body of your manuscript: (ie “Bewick et al. [Unpublished]”) as detailed online in our guide for authors

**Additional Editor Comments:**

Careful proofreading or professional editing of the language, rechecking of statistics, and clarity of figures are necessary.

Reviewers' comments:

Reviewer's Responses to Questions

**Comments to the Author**

1. Is the manuscript technically sound, and do the data support the conclusions?

Reviewer #1: Yes

Reviewer #2: Yes

2. Has the statistical analysis been performed appropriately and rigorously? 

Reviewer #1: Yes

Reviewer #2: No

3. Have the authors made all data underlying the findings in their manuscript fully available?

Reviewer #1: Yes

Reviewer #2: No

4. Is the manuscript presented in an intelligible fashion and written in standard English?

Reviewer #1: Yes

Reviewer #2: Yes

5. Review Comments to the Author

Reviewer #1: This manuscript investigates the crucial relationship between water stress, salicylic acid (SA) signaling, and chemical and structural defenses (Picea engelmannii). The authors provide compelling evidence that water stress alters the plant's hormonal balance, resulting in changes in the monoterpene profile and, most notably, suppression of the tree's ability to induce structural defenses (traumatic resin ducts). The manuscript is relevant and offers a basis for the observed correlation between droughts and increased insect outbreaks in the forest. The study is well-designed, and the findings are robust; however, the manuscript requires further in-depth and focused discussion, as well as improvements in methodological clarity to achieve high-impact publication rigor.

The most significant finding is the suppression of traumatic resin duct induction under chronic water stress conditions (Experiment 3). However, the manuscript should further explore the implications of the uncoupling of SA signaling observed between needles and stem phloem (Experiment 2). The correlation between increased SA and increased specific monoterpenes under water stress is a key finding, particularly in the stem phloem (Fig. 4).

The discussion should articulate the hypothesis that SA is modulating terpene biosynthesis as part of a protective response against drought-induced oxidative stress (ROS scavenging). Water stress leads to photosynthetic inhibition and the accumulation of reactive oxygen species (ROS).

The Methods section should be more detailed. The complete protocol for quantification of SA and monoterpenes (GC-MS/LC-MS specifications, column parameters, detection limits) should be fully described in the main body of the text, and not just in the captions or supplementary material. Correct obvious typos (e.g., "forest instects" for "forest insects"; "steam water potential" for "stem water potential").

In conclusion, this manuscript presents a valuable and insightful contribution to the understanding of how water stress compromises the resistance of conifers to herbivores, providing a solid hormonal basis for ecological vulnerability. Acceptance for publication is conditional upon revision of the text to intensify the mechanistic discussion, especially regarding SA-JA antagonism and the antioxidant function of SA-induced monoterpenes.

Reviewer #2: The study is scientifically sound and addresses an ecologically important topic linking drought stress, hormone signaling, and defensive chemistry in Picea engelmannii. However, several aspects of the methodology, clarity, statistical analysis, and structuring require revision before acceptance.

The study design is appropriate, the experiments are well-organized into three logical components, and the datasets generated (hormone profiles, monoterpenes, resin duct formation) are relevant to the stated hypotheses. However, improvements in methodological detail, statistical rigor, and interpretation are needed to fully confirm that the data justify the conclusions.

- Terms such as “apparency,” “resin duct density,” “basipetal signaling,” etc. should be briefly defined on first use for broader accessibility.

-some figure lack full clarity and could be expanded to briefly explain

- Lack of explicit controls in Experiment 3

- Using α = 0.10 for significance is unconventional.Must be justified or adjusted.

-Statistical assumptions (normality, variance homogeneity) are not mentioned.Without this, model validity is uncertain.

-ANOVA Model Structure Needs Detail. The phrase “watering treatment nested by time” is ambiguous.

-Effect Sizes Not Reported

- Randomization and blinding are not described. Greenhouse bench effects may exist.

- Some statements in the Discussion imply causal relationships not directly tested (e.g., SA antagonizing JA).Conclusions should be rewritten to remain strictly within the boundaries of the data.

Language:The manuscript is mostly intelligible and readable, but improving grammar, phrasing, and clarity will enhance the professional quality of the work. A careful proofreading or light professional editing is recommended.

6. PLOS authors have the option to publish the peer review history of their article (what does this mean?). If published, this will include your full peer review and any attached files.

Reviewer #1: No

Reviewer #2: **Yes:** Amisha S Amin

---

## [Author Response · Author response to Decision Letter 1]

11 Jan 2026

Reviewers' comments:

Reviewer 1:

The most significant finding is the suppression of traumatic resin duct induction under chronic water stress conditions (Experiment 3). However, the manuscript should further explore the implications of the uncoupling of SA signaling observed between needles and stem phloem (Experiment 2).

>We agree that this is an interesting angle, however, at the suggestion of R2 we have decided to refrain from extensively speculating on mechanisms that were beyond the scope of the present work. Accordingly, we have re-phrased this section to instead read “This could indicate that different environmental factors drive accumulation of SA in needle and stem tissues (Vanhatalo et al. 2015)…”, and no longer refer to a ‘decoupling’ as this was not directly tested or observed.

The discussion should articulate the hypothesis that SA is modulating terpene biosynthesis as part of a protective response against drought-induced oxidative stress (ROS scavenging). Water stress leads to photosynthetic inhibition and the accumulation of reactive oxygen species (ROS).

>We have added some text to address this idea, as suggested.

The Methods section should be more detailed. The complete protocol for quantification of SA and monoterpenes (GC-MS/LC-MS specifications, column parameters, detection limits) should be fully described in the main body of the text, and not just in the captions or supplementary material.

>We have included information in-text on the GC-MS temperature program. Column parameters and carrier gas flow rates are provided L231-239. Likewise, relevant instrument parameters for UPLC analyses are provided at L206-218. Additional detail on detection limits is now provided in the data acquisition section on L250-255.

We felt that detailed in-text discussion of solvent gradients and (Table S1) and especially listing of all monoterpene transition quantifier ions (Table S4) was simply too cumbersome to include in-text, but these details are transparently viewable in supplemental tables.

Correct obvious typos (e.g., "forest instects" for "forest insects"; "steam water potential" for "stem water potential").

>Corrected as suggested here.

Acceptance for publication is conditional upon revision of the text to intensify the mechanistic discussion, especially regarding SA-JA antagonism and the antioxidant function of SA-induced monoterpenes.

>We added relevant information along with citations, including a more detailed discussion of potential mechanistic roles of SA and monoterpene induction during drought stress. However, we have tried to be careful not to extend our discussion beyond the scope of what was observed and tested in the current studies as per the comments of R2 below. Accordingly, we now introduce ‘hormone crosstalk’ and ‘carbon starvation’ as potential competing hypotheses explaining a lack of resin duct formation in drought-stressed trees in response to MeJa.

Reviewer 2:

Terms such as “apparency,” “resin duct density,” “basipetal signaling,” etc. should be briefly defined on first use for broader accessibility.

>We now define ‘apparency’ (L425) at first use in the main text. The term ‘basipetal signaling’ could not be located in the text. ‘Resin duct density’ has been changed to ‘resin duct formation’ throughout to improve clarity.

Some figures lack full clarity and could be expanded to briefly explain

>Figure summaries have been added for each caption as per journal guidelines, which should help to improve the clarity of figures. We have also included some additional interpretation in figure captions where appropriate.

Lack of explicit controls in Experiment 3

>We are unclear on the meaning of ‘explicit controls’. Here the adjusted experimental variable is water level (n=2). It is already well established that application of MeJa induces traumatic resin ducts in spruces including the study species (e.g. Davis 2025, reference now included), so we did not consider it necessary to demonstrate that not applying MeJa does not result in resin duct production.

Reference:

Davis 2025. https://doi.org/10.1007/s10886-025-01665-4

Using α = 0.10 for significance is unconventional. Must be justified or adjusted.

>Several references have been added at this line to justify the use of 0.10 for interpreting statistical significance. This usage is increasingly common in ecological fields where variation can be extreme and study conditions may be impossible to identically replicate.

References:

Wasserstein et al. 2019: https://doi.org/10.1080/00031305.2019.1583913

Amrhein et al. 2019: https://doi.org/10.1038/d41586-019-00857-9

Statistical assumptions (normality, variance homogeneity) are not mentioned. Without this, model validity is uncertain.

>We have added a statement to this effect. We also now discuss the distribution of monoterpene data, and justify use of ANOVA from recent research.

Reference:

Blanca et al. 2017. https://doi.org/10.7334/psicothema2016.383

ANOVA Model Structure Needs Detail. The phrase “watering treatment nested by time” is ambiguous.

>This has been slightly revised to make it clearer that treatment effects are compared within each day, hence the term ‘nested within day’. To our knowledge, a ‘nested’ design is common statistical terminology (e.g., Schielzeth and Nakagawa 2012), and we are not sure how else to phrase this.

References:

Schielzeth and Nakagawa: https://doi.org/10.1111/j.2041-210x.2012.00251.x

Effect Sizes Not Reported

>For simple linear regressions or ANOVA models with only one factor (i.e., Figure 1), we now include coefficients of determination as an estimate of effect size.

For our nested two-way ANOVA model evaluating effects of SA concentration, watering treatment, and their interaction on monoterpene concentrations (Table 1), effect sizes can be generalized from the sum-of-squares column.

For our nominal logistic model (Figure 5), we now report the log-worth value for each model parameter in parenthetical statements as an estimate of parameter effect size.

Randomization and blinding are not described. Greenhouse bench effects may exist.

>Randomization and blind labeling are now described in the methods. All treatment positions were completely randomized on greenhouse benches.

Some statements in the Discussion imply causal relationships not directly tested (e.g., SA antagonizing JA).

>We agree, and refrain from stating this. Instead, we now refer to this as a possible explanation of why drought stress inhibited resin duct formation in response to MeJa, and provide other competing hypotheses as well (ie, carbon starvation).

Conclusions should be rewritten to remain strictly within the boundaries of the data.

>We have tried to be careful about this, while also providing some additional discussion as requested by R1.

Language: The manuscript is mostly intelligible and readable, but improving grammar, phrasing, and clarity will enhance the professional quality of the work. A careful proofreading or light professional editing is recommended.

>We have carefully proofread the ms and made edits as needed throughout to improve clarity where possible.

---

## [Decision Letter · Decision Letter 1]

28 Apr 2026

Drought stress induces salicylic acid accumulation, altering monoterpene profile and suppressing resin duct formation in Engelmann spruce

PONE-D-25-50354R1

Dear Dr. Davis,

We’re pleased to inform you that your manuscript has been judged scientifically suitable for publication and will be formally accepted for publication once it meets all outstanding technical requirements.

Kind regards,

Mozaniel Santana de Oliveira, Ph.D

Academic Editor

PLOS One

Additional Editor Comments (optional):

Reviewers' comments:

Reviewer's Responses to Questions

**Comments to the Author**

1. If the authors have adequately addressed your comments raised in a previous round of review and you feel that this manuscript is now acceptable for publication, you may indicate that here to bypass the “Comments to the Author” section, enter your conflict of interest statement in the “Confidential to Editor” section, and submit your "Accept" recommendation.

Reviewer #1: (No Response)

Reviewer #2: All comments have been addressed

Reviewer #3: All comments have been addressed

2. Is the manuscript technically sound, and do the data support the conclusions?

Reviewer #1: (No Response)

Reviewer #2: Yes

Reviewer #3: Yes

3. Has the statistical analysis been performed appropriately and rigorously? 

Reviewer #1: (No Response)

Reviewer #2: Yes

Reviewer #3: Yes

4. Have the authors made all data underlying the findings in their manuscript fully available?

Reviewer #1: (No Response)

Reviewer #2: Yes

Reviewer #3: Yes

5. Is the manuscript presented in an intelligible fashion and written in standard English?

Reviewer #1: (No Response)

Reviewer #2: Yes

Reviewer #3: Yes

6. Review Comments to the Author

Reviewer #1: (No Response)

Reviewer #2: The revised manuscript shows clear improvement compared with the previous version, and the authors have made a commendable effort to address earlier reviewer comments. The study presents a technically sound investigation, and the experimental design, data presentation, and overall structure of the manuscript are generally appropriate. The findings contribute useful insights into plant physiological and biochemical responses under drought conditions and their potential implications for plant defense mechanisms.

The statistical approaches applied are generally suitable for the study design.

A few minor issues remain that could further strengthen the manuscript. In particular, some interpretations in the discussion should be framed more cautiously to avoid implying causal relationships where the study primarily demonstrates associations. Additionally, clarification of certain statistical details and explicit reporting of effect sizes would improve methodological transparency.

Overall, the manuscript represents a valuable contribution to the field and, with minor clarifications and careful interpretation of the results, it is suitable for publication.

Reviewer #3: The authors have adequately addressed all prior reviewer comments, and the manuscript is now technically sound, clearly written, and appropriately interpreted. The experimental design and analyses are rigoro and resin duct formation. All underlying data are fully available in compliance with the PLOS Data Policy.

7. PLOS authors have the option to publish the peer review history of their article (what does this mean?). If published, this will include your full peer review and any attached files.

Reviewer #1: No

Reviewer #2: **Yes:** Amisha S Amin

Reviewer #3: No

---

## [Editor Report · Acceptance letter]

PONE-D-25-50354R1

PLOS One

Dear Dr. Davis,

I'm pleased to inform you that your manuscript has been deemed suitable for publication in PLOS One. Congratulations! Your manuscript is now being handed over to our production team.

Kind regards,

on behalf of

Dr. Mozaniel Santana de Oliveira

Academic Editor

PLOS One